# FROM GRADIENT ATTACKS TO DATA POISONING

## ABSTRACT

Security concerns around gradient attacks - in which an adversary can inject a maliciously crafted gradient during the training - have long been studied in distributed learning due to their proven harmfulness and the difficulty to defend against. These attacks however have been argued to affect far less systems than data poisoning. In the latter, an attacker's power is reduced to only being able to inject data points to training sets, via e.g. legitimate-looking participation in an online service, or participation in a collaborative or open-sourced dataset. Even though an equivalence between the two attack modalities have been showed in convex settings (regression), this apparent difference in the attackers' power raises the question of whether the harm made by gradient attacks can be replicated by data poisoning ones in non-convex settings. In this paper, we show that data poisoning can sometimes mimic gradient availability attacks in a more practical deep learning setting. While data poisoning have mainly been used to perform targeted or backdoor attacks, we show that by borrowing a threat model to gradient attacks, we can successfully perform a data poisoning availability attack.

## 1 INTRODUCTION

In Federated Learning, workers only communicate the gradients computed on their own data. One malicious worker can, alone, steer the training procedure with a gradient attack, where it can send an arbitrary gradient as showed in Blanchard et al. (2017). These attacks can severely degrade the global performances of the model (availability attacks). However, most machine learning models are trained on the data points directly. While Machine Learning practitioners tend to trust their training procedures, their data, and resulting models (Hoang et al., 2021), data poisoning, which consists in tampering with training samples to induce a certain bias or behaviour in a machine learning model, has proven to be a realistic security threat (Carlini et al., 2023) that can throw algorithms off (Biggio et al., 2013; Geiping et al., 2021).

As stated in the abstract of Gebru et al. (2021) : *"The machine learning community currently has no standardized process for documenting datasets, which can lead to severe consequences in high-stakes domains"*. This is particularly true as they scaled to web-scrapped datasets (Luccioni & Viviano, 2021; Schuhmann et al., 2022). Because of such scale, it is difficult to comprehend the extent of contamination they could realistically endure. Although Carlini et al. (2023) shows that they manage to poison at most 1% of a few publicly available datasets with a $10k budget, a much higher upper bound can be advanced, and higher levels of contamination are not to overlook either. Varol et al. (2017) claimed that bots represented between 9% and 15% of Twitter's active users (which is more than Twitter's CEO's claim of 5% Agrawal (2022)). We could expect the ratio of stealthy malevolent agents to be somewhat similar (if not higher). And since Xie et al. (2022) shows that a subtle substitution of one word with a synonymous can be enough to perform an attack, even a strict moderation cannot hope to prevent these attacks from stealthy adversarial accounts. On Reddit, although there is no relevant estimation of the number of fake accounts, creating an account is quick and easy. So much that the notion of "throw away account" is common to post anonymously (MEOWTheKitty18, 2022). We thus expect the ratio of potential malevolent accounts and potentially poisonous contents there to be much higher, so much that it has already influenced Machine Learning algorithms. Rumbelow & mwatkins (2023) shows that a few redditors artificially inflated their online presence (by posting over 160k posts on the "r/counting" subreddit on which people simply count) on the platform so much that they managed to have a dedicated token in GPT-3's tokenizer. According to Charon & Vilmer (2021), "armies" as large as 2 million full time and

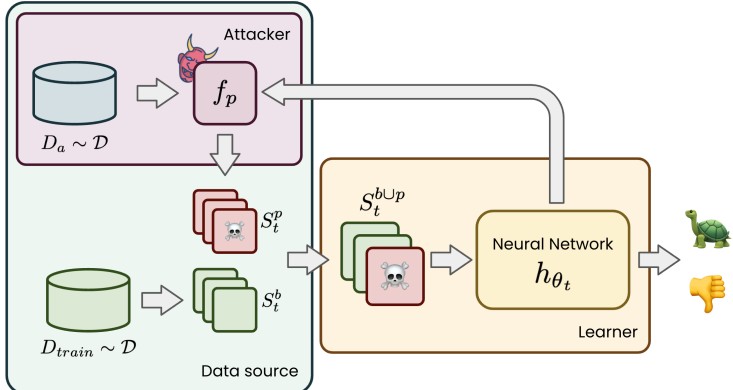

Figure 1: Threat model for data poisoning. The attacker does not have access to the minibatch $S_t^b$ but estimates it with an auxiliary dataset $D_a$ to craft $S_t^p$ the set of data poisons. Both the minibatch and the poisons set are gathered into $S_t^{b\cup p}$. The attacker's goal is either to slow down the training or prevent it.

20 million part time individuals could be conducing campaigns on social media. Datasets could already be poisoned without easily detectable action: social media, collaborative datasets such as CommonVoice (Ardila et al., 2020) or Wikipedia (Carlini et al., 2023), or even intruding the websites hosting trusted datasets.

What is, then, the extent of harm that can be caused by data poisoning on deep learning models? A partial answer by Farhadkhani et al. (2022) shows an equivalence between gradient attacks and data poisoning in convex settings. Such result has yet to be extended to deep neural networks in non-convex settings. In this work, we borrow a threat model from gradient attacks litterature and allow an attacker to send a few adversarially crafted data points at each iteration during the training process. This threat model acts like a worst-case scenario from a data poisoning perspective as attackers usually only craft their poison once and put them in the dataset among other points. We formulate our data poisoning procedure as an optimization problem constrained in the input space which aims at producing data poisons whose resulting gradient is as similar as possible as a given gradient attack. As such, we demonstrate that data poisoning can mimic gradient attacks and successfully perform a training time attack which slows down the training or an availability attack which degrades the model's global performances.

CONTRIBUTIONS

- We propose a new data poisoning threat model inspired from gradient attacks.
- We design new data poisoning attacks by mimicking gradient attacks.
- We experimentally demonstrate these data poisoning procedures can produce training time attacks and availability attacks on neural networks trained end-to-end.

## 2   BACKGROUND

Our work builds on existing knowledge of the harmfulness of both gradient attacks and data poisoning. There are several types of attacks: targeted toward certain examples (*integrity attack*), untargeted to compromise the general performances of the model (*availability attack*), triggerless when the adversary can only influence the training phase of the model, backdoor when it can also intervene at inference time...

**Gradient Attacks**   are applicable when the learner uses gradients sent by the worker, such as in Federated Learning. There is naively no way to determine if a gradient is legitimate or not. So much that a malevolent (Lamport et al., 1982) worker can send **any** gradient vector. Lemma 1 in Blanchard et al. (2017) shows that AVERAGING in Stochastic Gradient Descent SGD can be

devastated by only a single malevolent worker and that the way by which gradients are aggregated can be a defense mechanism since learning under gradient attack can be reduced to robust mean estimation (Yin et al., 2021). However, with better defense mechanisms came better attacks (Baruch et al., 2019; El-Mhamdi et al., 2018). Even stronger, Theorem 2 in El-Mhamdi et al. (2023) shows the impossibility of robust mean estimation, and therefore of robust distributed learning, below a certain threshold that depends on data heterogeneity and model size. We leverage working gradient attacks in Federated Learning settings and apply them in Centralized Learning settings where the use of defense mechanisms are yet to be the norm (Bouhata et al., 2022).

**Data Poisoning** manipulate the training data of ML models with the goal of influencing their behavior. If Farhadkhani et al. (2022) shows an equivalence with gradient attacks in the convex case, it falls short on neural networks and is limited to a frozen network with only the last layer trainable. This boils down to a linear regression and requires an impractical knowledge [1] on the trained model. A gradient attack can immediately be derived from a data poisoning (the attacker sends the gradients associated with the poisons), but the contrary may not be possible as the gradient attack may fall outside of the image space of the gradient operator. Our experiments show the existence of poisons derived from gradient attacks which achieve a similar impact on the model in a similar threat model, i.e. when the attacker crafts a poison at each update iteration of the model. Several ways of generating poisons for an availability attack exist: label flipping (Shejwalkar et al., 2021), generative models (Muñoz-González et al., 2019; Zhao & Lao, 2022), or gradient-based approaches. The latter allows us to finely control the resulting gradient on the poisonous points instead of relying on another proxy. Even though Shejwalkar et al. (2021) consider data poisoning to be of limited harm and gradient-based approach to be too computationally intensive, several work showed that such poisons can both be stealth and effective as in targeted clean-label attacks (Shafahi et al., 2018). These poisons are obtained as a solution to a a bi-level optimization problem in which the inner problem is the poisoning objective, and the outer problem is the learning one (Geiping et al., 2021). If direct solutions to said bi-level problem have been proposed for certain learning algorithms (Demontis et al., 2019), it is intractable for deep neural networks for which no closed form solution is known. Existing approaches are computationally expensive such as Muñoz-González et al. (2017) which inverts the training process and requires access to the dataset. We circumvent this difficulty by allowing the attacker to intervene at each iteration, only requiring to steer a single model update at a time, which lightens the computational burden. On the other hand, attacks on the training time have been understudied compared to other forms of attacks. Shumailov et al. (2021) proposes such attack, however, their adversary is allowed to order the data that are seen by the model. Such threat model cannot easily be adapted to other learning settings. In comparison, our threat model does not assume any particular learning setting.

**Defenses** To counter data poisoning attacks, several techniques were proposed such as data sanitization (Steinhardt et al., 2017), data augmentation (Borgnia et al., 2020), bagging (Wang et al., 2022) or pruning and fine-tuning (Liu et al., 2018). However, the effectiveness of such defenses rely on strong assumptions such as the learner having access to a clean dataset, on the convexity of the loss w.r.t. the model's parameters, or on the ability to train a very large number of models. And attackers can still find ways to break these defenses (Koh et al., 2021). Even though defending against data poisoning might be impossible, adopting defense mechanisms to make the attacker's job harder in a "swiss cheese"-like model is not to overlook.

## 3 SETTING

### 3.1 LEARNING SETTING

We consider a classification task where the model is trained on a dataset $D_{train} = \{(x_i, y_i)\}_{i=1}^n$ sampled from a distribution $\mathcal{D}$ over $\mathcal{X} \times \mathcal{Y}$. The learner trains a model $h_\theta$ parametrized by $\theta$ with a gradient-based minibatched optimization algorithm on the loss function $L$. Its goal is to achieve the

---

[1] knowing the embeddings of the input w.r.t. the frozen layers

lowest loss on a heldout test set $D_{test}$, formulated as the following optimization problem:

$$\theta^* \in \arg\min_{\theta \in \Theta} \frac{1}{n_{test}} \sum_{(x,y) \in D_{test}} L(h_\theta(x), y)$$

At each iteration $t$, a minibatch $S_t^b = \{(x_t^i, y_t^i)\}_{i=1}^{n_b}$ of size $n_b$ is sampled from $D_{train}$. The parameters are updated using an UPDATE algorithm which rely on an aggregation function AGG: $\theta_{t+1} = \text{UPDATE}(\theta_t, S_t^b, \text{AGG})$. In the common SGD setting, the update algorithm is SGD and the aggregation function is AVERAGING. The choice we make is important since in Federated Learning litterature, the update algorithm and aggregation rule are both considered as potential defense mechanisms. In our experiments, we will consider an image classification task on the CIFAR10 dataset using the SGD and ADAM update algorithms and the AVERAGING and MULTIKRUM Blanchard et al. (2017) aggregation rules. MULTIKRUM$_{f<0.5}$ is a robust aggregation rule and is defined for a set of vectors $\{v_i\}_{i=1}^n$ as the average of the $n(1-f)-2$ vectors minimizing the score function $s(i) = \sum_{i \to j} \|v_i - v_j\|^2$ with $i \to j$ the indices of the $n(1-f)-2$ closest vectors to $v_i$.

### 3.2 THREAT MODEL

We consider a threat model where the attacker:

- has access to the weights of the model $\theta_t$, the update algorithm UPDATE and aggregation rule AGG as in Farhadkhani et al. (2022); Blanchard et al. (2017); Yin et al. (2021); Shafahi et al. (2018). This can be the case if the attacker has gained read access to the system storing the model.

- has access to an auxiliary dataset $D_a \sim \mathcal{D}$ that is not the training set. This can be the case if the attacker has knowledge of the data used to train the model and constitutes a similar dataset, or if the attacker has gained read access to the system storing the training data and extract a part of them, or else in an online learning setting, if the attacker also sample points from the same stream of data as the learner. This was also a working assumption in Blanchard et al. (2017); Farhadkhani et al. (2022) and lie inbetween full knowledge of the training set Muñoz-González et al. (2017) and knowledge of a few targets only Geiping et al. (2021).

- has the ability to add a set of crafted data $S_t^p = (x_{t,i}^p, y_{t,i}^p)_{i=1}^{n_p}$ to the minibatch $S_t^b$ at each iteration $t$ up to a level $\alpha$ of contamination ($\frac{|S_t^p|}{|S_t^{b \cup p}|} = \alpha$). This assumption is borrowed from gradient attacks (Blanchard et al., 2017) and can be seen as a worst case scenario. We considered that pulling out an availability attack required slightly more interaction with the model than traditional data poisoning assumptions where the poisoner craft its poisons once and for all (Geiping et al., 2021; Shafahi et al., 2018; Muñoz-González et al., 2019). But it can also be a working assumption in an online learning setting where the attacker is a regular user of the system and can add data to the stream of data (as in a Sybil attack). This is somewhat implicitly assumed in Xie et al. (2022).

- constrains itself with only crafting poisons that are in a feasible domain $S_t^p \in \mathcal{F}^{n_p} = (\mathcal{F}_\mathcal{X} \times \mathcal{F}_\mathcal{Y})^{n_p}$, depending on the task and data structure. For instance, for an image classification task on CIFAR10, where each image's $32 \times 32$ pixels are coded on 3 bytes and labels are between 1 and 10, we set $\mathcal{F}$ to be the set of possible such sized image and labels: $\mathcal{F} = [0..255]^{32 \times 32 \times 3} \times [1..10]$.

The attacker's goal is to perform an **availability attack:** degrading the learner's performances as much as possible. A weaker version of such attack would be to slow down the learning procedure in a **training time attack** (see Figure 1).

## 4 METHOD

To achieve its goal, at each iteration $t$, the attacker estimates the clean averaged gradient over the minibatch: $g_t^b = \frac{1}{n_b} \sum_{g \in G_t^b} g$ with $G_t^b = \{g_{t,i}^b\}_{i=1}^{n_b} = \{\nabla_{\theta_t} L(h_{\theta_t}(x), y)\}_{(x,y) \in S_t^b}$ the set of clean sample gradients over the minibatch. Since the attacker cannot access the minibatch, it uses the

clean gradients over the auxiliary dataset $G_t^a$ as a surrogate to estimate the clean averaged gradient over the minibatch: $g_t^a = \frac{1}{n_a} \sum_{g \in G_t^a} g$ with $G_t^a = \{g_{t,i}^a\}_{i=1}^{n_a} = \{\nabla_{\theta_t} L(h_{\theta_t}(x), y)\}_{(x,y) \in D_a}$ and $\mathbb{E}_{G_t^a}[g] = \mathbb{E}_{G_t^b}[g]$. The attacker can also compute the sample gradients $G_t^p$ over the poisonous points: $g_t^p = \frac{1}{n_p} \sum_{g \in G_t^p} g$ with $G_t^p = \{g_{t,i}^p\}_{i=1}^{n_p} = \{\nabla_{\theta_t} L(h_{\theta_t}(x), y)\}_{(x,y) \in S_t^p}$. Similarly, the averaged gradient over the poisoned minibatch is estimated with the weighted averaged gradient over the auxiliary dataset and poisoned points: $g_t^{b \cup p} = \frac{1}{n_b + n_p} (n_b g_t^b + n_p g_t^p)$, $g_t^{a \cup p} = \frac{1}{n_b + n_p} (n_b g_t^a + n_p g_t^p)$, and $\mathbb{E}_{G_t^a}[g_t^{a \cup p}] = \mathbb{E}_{G_t^b}[g_t^{b \cup p}]$. We then formulate our data poisoning as an optimization problem where the attacker aims to minimize a poisoning function $f_p$ over a feasible domain $\mathcal{F}$:

$$S_t^p \in \arg \min_{S \in \mathcal{F}^{n_p}} f_p(h_\theta, D_{aux}, S) \tag{1}$$

One way such attacker could perform a data poisoning would thus be to try and obtain a targeted gradient with the poisoned surrogate gradient. Table 1 details the formulas used to determine the gradient attacks and their equivalent poisoning functions in the data poisoning case.

### 4.1 GRADIENT ATTACKS

We suggest several gradient attacks to perform an availability attack.

- **Gradient Ascent** (Blanchard et al., 2017): the attacker sends a gradient such that the total corrupted gradient is anti-collinear with the clean gradient. Which would provoke a gradient ascent step, with the SGD update rule and $\lambda \in \mathbb{R}$:

$$\mathbb{E}_{G_t^b}[\theta_{t+1}] = \theta_t - \eta \mathbb{E}_{G_t^b}[g_t^{b \cup p}] = \theta_t - \eta \mathbb{E}_{G_t^a}[g_t^{a \cup p}] = \theta_t + \eta \lambda \times \mathbb{E}_{G_t^b}[g_t^b]$$

- **Orthogonal Gradient**: the attacker sends a gradient such that the total corrupted gradient is orthogonal to the clean gradient. This should make the training stall.

- **Little is Enough** (Baruch et al., 2019): the attacker sends the mean gradient deviated by a strategically chosen amount times the coordinate-wise standard deviation of the gradients, with $\sigma[j] = \sqrt{Var(\{g[j]\}_{g \in G_t^a})}$.

$$g_t^p = g_t^a - z^{max} \times \sigma$$
$$with \ z^{max} \in \arg \max_{z \in \mathbb{R}_+^*} \left\| \text{AGG}(G_t^a \cup \{g_t^a - z\sigma\}_{i=1}^{n_p}) - \text{AGG}(G_t^a) \right\|$$

This attack takes the aggregation rule into consideration and has been shown to be effective against MULTIKRUM when the working conditions of the latter are not met.

### 4.2 DATA POISONING

Data poisoning is allegedly harder as it constitutes a far more constrained problem than gradient attacks (in which the attacker can send any gradient). We want the data poisons to be feasible, i.e. to be in a constrained domain $\mathcal{F}$. Characterizing the image space of the gradient operator on the loss function $L$ is a difficult task. Together with the imposed constraints on feasibility, finding data poisons which exactly reproduce the gradient attack might be impossible. We reformulate gradient attacks equations as optimization problems of carefully chosen poisoning functions $f_p$ in the feasible input domain $\mathcal{F}$. We show in our experiments that an attacker can still produce data poisons which have significant impact on the training procedure. The attacker does not have access to the minibatch and uses the auxiliary dataset as a surrogate to estimate any desired quantities. Instead of a gradient $g_t^p$, it sends a set of poisonous points $S_t^p$ to the minibatch. The poisoning is performed by solving the optimization problem (1) with the poisoning function as specified in Table 1. Poisons are iteratively updated using the `Adam` optimizer and are projected on the feasible set $\mathcal{F}$ at each iteration of the poisoning optimization by clipping.

## 5 EXPERIMENTS

### 5.1 EXPERIMENTAL SETUP

**Model & dataset** As a matter of proof of concept, we demonstrate our poisoning procedure on a custom convolutional neural network (described in Table 5 in Appendix) and on Vision Transformers

Table 1: Gradient attacks formula and their equivalent data poisoning objective functions to be optimized. $\theta$ represents the model's weight at the current iteration, and $g^p, g^a, g^{a\cup p}$ are respectively the resulting gradients computed on the data poisons, on the auxiliary dataset, and the weighted average between them.

| | **Gradient attack** $g^p \in \mathbb{R}^d$ s.t. | **Our data poisoning attack** $S^p \in \arg\min_{S\in\mathcal{F}^{n_p}} \dots$ |
|---|---|---|
| Gradient Ascent | $Sim_{cos}(g^{a\cup p}, g^a) = -1$ | $Sim_{cos}(g^{a\cup p}, g^a)$ |
| Orthogonal Gradient | $\langle g^{a\cup p}, g^a \rangle = 0$ | $\|Sim_{cos}(g^{a\cup p}, g^a)\|^2$ |
| Little is Enough | $g^p = g^a - z^{max} \times \sigma$ | $\|g^p - g^a + z^{max}\sigma\|^2$ |

(`ViT-tiny` models with patch size 8) models on the CIFAR10 dataset splitted in a partition of training, validation, and auxiliary datasets.We use different optimization algorithms and aggregation rules: SGD & AVERAGING, ADAM & AVERAGING, SGD & MULTIKRUM (with different levels of data truncation $f \in \{0.1, 0.2, 0.4\}$). Since MULTIKRUM is used as a defense mechanism, we expect it to be robust to the attacks, except for the Little is Enough attack which is specifically designed to circumvent it.

**Baseline** In every setting, the learning rate is fixed to the value were the learner achieves the best performances without any poisoning to set a baseline for the performances of the model. We then measure the decay in performances caused by an attack. Each setting of this experiment is run 4 times for better statistical significance. Each run has a different model and poisons initialization, and dataset split. Full results can be seen in Appendix. The $n_p$ crafted poisons are added to the minibatch of size $n_b$ at each iteration so that $\frac{n_p}{n_b+n_p} = \alpha$.

## 5.2 RESULTS

**Gradient attacks** To ensure that gradient attacks can have an impact in our setting, we perform them as a sanity check. They will also serve as toplines for the corresponding data poisoning procedures. Table 4 in Appendix shows that the gradient attacks perform well, making the model diverge, but also that the MULTIKRUM aggregation rule does act as a defense mechanism for levels of contamination lower than its tolerance parameter $f$. However, as expected, this defense is well circumvented by the Little is Enough attack.

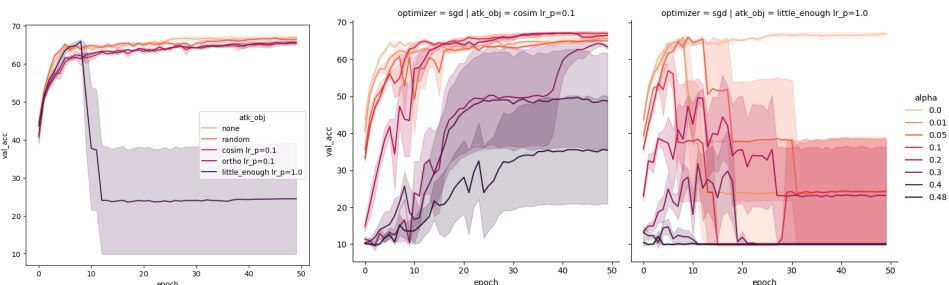

(a) Comparison of different attacks at $\alpha = 0.01$.

(b) Comparison of different levels of contamination for the Gradient Ascent & Little is Enough attacks.

Figure 2: Validation accuracies during training in the SGD & AVERAGING setting under different attacks and different level $\alpha$ of contamination. Error bars represent the standard error.

**Overview of the data poisoning** We now perform the equivalent data poisonings. The observed effects can be graduated from a slowdown of the training procedure to its complete halt and up to degrading the performances down to random levels. Figure 2a compares the different attacks and a random data poisoning (uniformly sampled in $[0, 1]$) at $\alpha = 0.01$ with its baseline counterpart. While the Gradient Ascent and Orthogonal attacks behave similarly and only slightly slow down the training at most, Little is Enough attack appears able to strongly degrade the performances,

even at such low level of contamination. Figure 2b shows for the Gradient Ascent and Little is Enough attacks that the higher the level of contamination, the stronger the effect: validation accuracy increases slower or plummets earlier in the training.

**Availability attacks**   To fairly compare each setting under that attack, each model has been evaluated on the test set at the point where it achieved the best validation accuracy. Figure 3 shows that for the SGD & AVERAGING learner setting under the Little is Enough attack, $\boldsymbol{\alpha = 0.01}$ **is enough to significantly degrade the model's performances**. Among the 4 runs, 3 ended up diverging while every poison was **in the feasible set** $\mathcal{F}$. Figure 7 in the Appendix shows 50 of the first 500 poisons crafted by the attacker in one of the diverging run. Similarly, in the setting SGD & MULTIKRUM$_{f=0.1}$, the attacker performing the Little is Enough attack with contamination level $\boldsymbol{\alpha = 0.05}$ **is able to find data poisons that circumvent the robust aggregation rule** and drastically reduce the model's performances.

**Training time attacks**   While the gradient attacks we propose successfully perform availability attacks, the corresponding data poisoning might not totally succeed. Instead, it can find a solution that perturbs the total gradient enough to slow down convergence, but not enough to completely halt it. Figure 6 in Appendix illustrates this.

**Comparaison of update rules.**   We compare different update rules and aggregation functions under the same attack. Figure 3 shows that our data poisoning procedure failed at performing an availability attack against the ADAM update rule. However, the training time attack can still be observed. Table 2 shows that higher level of contamination lead to lower best validation accuracies and a higher number of epochs (hence a longer time) to reach it.

| $\alpha$ | GA | OG | LIE |
|---|---|---|---|
| 0.00 | 4 (67.1) | 4 (67.1) | 4 (67.1) |
| 0.01 | 5 (66.4) | 5 (66.4) | 6 (66.3) |
| 0.05 | 7 (66.5) | 7 (65.3) | 7 (66.3) |
| 0.10 | 19 (63.6) | 14 (63.9) | 10 (65.7) |
| 0.20 | 34 (61.2) | 17 (62.5) | 22 (62.1) |
| 0.30 | 19 (63.0) | 20 (62.0) | 23 (60.4) |
| 0.40 | 34 (63.5) | 23 (60.7) | 9 (36.5) |
| 0.48 | 13 (64.0) | 13 (65.0) | 48 (36.0) |

Table 2: Epoch at which the best val accuracy is reached (val accuracies in parenthesis) for the CNN model with ADAM optimizer under different data poisoning attacks. Higher levels of contamination display slower training and lower performances.

**Data poisoning against a standard robust aggregation rule.**   Table 3 shows that with the MULTIKRUM$_{f=0.1}$ aggregation rule which filter gradients, the attacker still manages to have some of them to be selected. For $\alpha > f$, the aggregation rule do not play a defensive role anymore. On the other hand, for $\alpha$ below this point, the Little is Enough attack displays significantly higher selection rates than the other attacks. This means that the attacker manages to produce data poisons whose gradients deceive MULTIKRUM$_{f=0.1}$, similarly to the gradient version of the attack which is particularly designed for this purpose. However Figure 3 shows that a higher selection rate does not necessarily means success of attack. Even if the attacker sometimes manages to successfully attack the model, MULTIKRUM overall enhances the robustness of the model while slightly degradiong its performances.

**Influence of the feasible set**   As the feasible set $\mathcal{F}_{\mathcal{X}}$ change, the attacker will only be allowed to converge (or not) towards different poisons. We compare three increasingly restrictive feasible sets to determine their influence on the success of the attack:

- Constraint-free set: the feasible set is simply the input domain $\mathcal{F}_{\mathcal{X}\text{free}} = \mathbb{R}^{H \times W \times 3}$;
- Image-encoding set: the feasible set ensures that the poisons respect the same encoding as clean data $\mathcal{F}_{\mathcal{X}\text{img}} = 2^{8^{H \times W \times 3}}$;

Table 3: Poison selection rates in the SGD & MULTIKRUM$_{f=0.1}$ averaged over all runs and all epochs. Each row corresponds to a different attack. Standard deviations in parenthesis.

|  | $\alpha$ | | | |
|---|---|---|---|---|
|  | **0.01** | **0.05** | **0.1** | **0.2** |
| Gradient ascent | 0.0 | 2e−4 | 1.5e−3 | **0.91** |
|  | (0.0) | (4e−4) | (2.4e−3) | (0.08) |
| Orthogonal | 0.0 | 3e−4 | 0.18 | 0.85 |
|  | (0.0) | (1e−3) | (0.1) | (0.05) |
| Little is Enough | **0.92** | **0.91** | **0.97** | 0.87 |
|  | (0.07) | (0.13) | (0.02) | (0.06) |

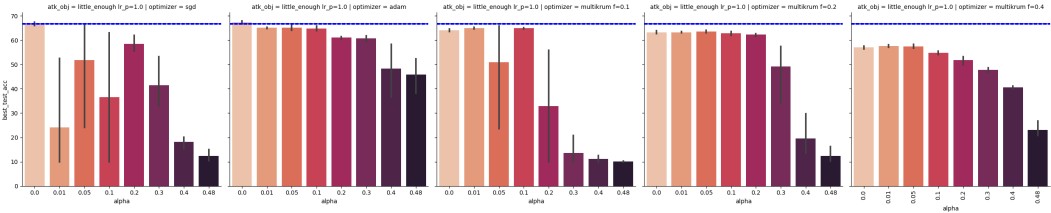

Figure 3: Test accuracy of the CNN model which achieved the best validation accuracy under the Little is Enough attack. Each column represents a different setting of update function and aggregation rule. The blue line is the test accuracy obtained with (SGD, AVERAGING) setting and no attack. Error bars are the standard errors.

- Neighborhood set: is a subset of the previous one and is composed of all the images that are at a L1 norm of at most $\epsilon = \frac{32}{255}$, $\mathcal{F}_{\mathcal{X}\mathrm{nei}} = \{x \in \mathcal{F}_{\mathcal{X}\mathrm{img}} / \exists x_a \in D_a s.t. \|x - x_a\|_1 \leq \epsilon\}$

Figure 4 shows that the more constrained the feasible set, the less effective the resulting attack. It is to note however that early stopping helps in limiting the effects of the attack (see Figure 8 in Appendix).

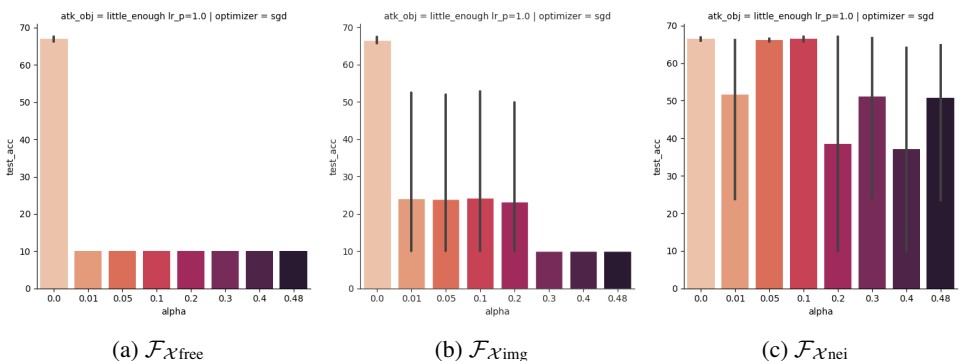

(a) $\mathcal{F}_{\mathcal{X}\mathrm{free}}$  (b) $\mathcal{F}_{\mathcal{X}\mathrm{img}}$  (c) $\mathcal{F}_{\mathcal{X}\mathrm{nei}}$

Figure 4: Final test accuracies for the SGD; AVERAGING setting under the Little is Enough attack for different feasible sets. Error bars are the standard errors.

**Neural Network architecture** We perform the same attacks on the same learning pipelines but replace the CNN model with a Visual Transformer (ViT) tiny with patch size 8. Since ViT are predominantly trained using ADAM-like approaches, we report the efficiency of our attacks using it. Figure 5 shows that ViT are also vulnerable to our data poisoning. Although with far less success and with higher levels of contamination required.

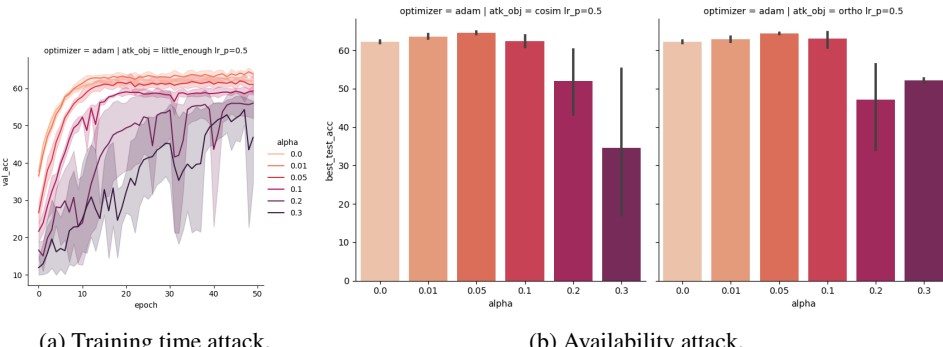

(a) Training time attack.  (b) Availability attack.

Figure 5: Visual Transformer (ViT) tiny with patch size 8 under different attacks. Little is Enough performs a training time attack whereas Gradient Ascent and Orthogonal Gradient are able to perform an availability attack for high enough contamination levels $\alpha$.

## 6 DISCUSSION

In this work, we show that in training settings involving deep neural networks, which are not restricted to convex cases as in (Farhadkhani et al., 2022), data poisoning can be crafted to mimic gradient attacks. Our threat model uses several assumptions that should be further explored in future work:

- The role of the auxiliary dataset. In our experiments, it appeared that the size of the auxiliary dataset played only a small part in the success of the attack above a sufficient size. But what if the auxiliary dataset is no longer sampled from the same distribution as the training set? What if the attacker cannot have access to an auxiliary dataset and estimates the mean clean gradient by observing the successive updates of the model?

- The necessity to access the trained model's weights. Exploring whether or not the attack can still be performed when the attacker estimates the victim model's with a surrogate model would open the threat model to a wider variety of cases.

- The role of the feasible set. Works on clean label poisoning (Geiping et al., 2021; Shafahi et al., 2018) showed that it is possible to design data poisons which decieve human annotators. Future work should consider exploring more constraining feasible sets for the attacker.

- Finally, our threat model acts as a worst case scenario, future work should explore data poisoning availability attacks in a more common threat model, i.e. limiting the number of times the attacker can craft its poisons.

Although the Little is Enough attack shows working cases of data poisoning on the model availability, the other attacks only manage to slow down the training at best. We consider this to still be an attack not to overlook, considering that companies might afford to have their costly trainings slowed down.

Overall, our attack does not consistently manage to craft an effective data poisoning for an availability attack. However, we argue that our work shows how a data poisoning mimicking a gradient attack *is possible*, even in the case of neural networks. We chose to solve the poisoning optimization problem with gradient-based approaches which showed to be computationally intense, limiting us in experimenting with larger models and datasets. Stronger or less computationally expensive data poisoning approach (that may not rely on gradient attacks) can be experimented to enhance the attack success rate.

## 7 CONCLUSION

This work shows that data poisoning can mimic gradient attacks and have a similarly devastating impact on neural networks trained end-to-end. While the tested defense mechanisms for the gradient

case appear to generalize to the data poisoning, it has been shown that data poisoning can bypass certain defense mechanisms Koh et al. (2021) and that robust mean estimation only work up to a certain point El-Mhamdi et al. (2023). This leaves room for potentially devastating data poisoning that can bypass defenses while mimicking an unstoppable gradient attack. Although the success rate of our attack was relatively low, the possibility of mimicking gradient attacks with feasible data poisoning ones should motivate further research in defense mechanisms and lower bounds to better assess the safety of machine learning algorithms in the presence of unreliable data.

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

# A  GRADIENT ATTACKS

Table 4: Sanity check with gradient attacks. Best validation accuracy under different attacks with different update rules and different aggregation functions. A high validation accuracy (colored in apricot) indicates a failed attack. A low validation accuracy (colored in pale green) indicates a successful attack.

| UPDATE; AGG | Attack | $\alpha$ | | | | | | |
|---|---|---|---|---|---|---|---|---|
| | | 0.01 | 0.05 | 0.10 | 0.20 | 0.30 | 0.40 | 0.48 |
| ADAM; AVERAGING | GA | 10.0 | 9.8 | 10.0 | 10.0 | 10.0 | 10.1 | 10.0 |
| | OG | 9.9 | 9.9 | 10.1 | 10.1 | 9.8 | 10.0 | 10.1 |
| | LIE | 10.2 | 10.3 | 10.1 | 10.2 | 10.1 | 10.1 | 10.0 |
| SGD; AVERAGING | GA | 10.1 | 10.1 | 10.1 | 9.9 | 10.0 | 10.0 | 10.0 |
| | OG | 10.2 | 10.2 | 10.0 | 9.9 | 10.1 | 10.1 | 10.2 |
| | LIE | 10.2 | 9.9 | 10.2 | 9.9 | 9.9 | 10.0 | 10.0 |
| SGD; MULTIKRUM$_{f=0.1}$ | GA | 65.1 | 63.2 | 33.7 | 10.1 | 10.1 | 10.2 | 10.2 |
| | OG | 65.1 | 65.3 | 65.9 | 10.0 | 10.1 | 10.2 | 9.9 |
| | LIE | 41.7 | 15.0 | 10.4 | 9.9 | 10.0 | 9.9 | 10.1 |

# B  DATA POISONING

Table 5: Architecture of the 1.6M parameters convolutional neural network used for our experiments.

| Layer | # of channels | kernel | stride |
|---|---|---|---|
| Conv2d | 32 | $5 \times 5$ | 2 |
| ReLU | | | |
| Conv2d | 64 | $5 \times 5$ | 2 |
| ReLU | | | |
| Linear | 512 | | |
| ReLU | | | |
| Linear | 64 | | |
| ReLU | | | |
| Linear | 10 | | |

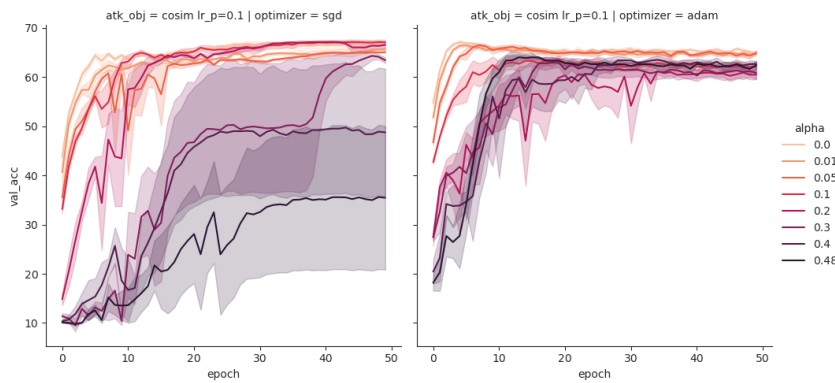

Figure 6: Validation accuracies of the CNN during training with the SGD and ADAM update rule with the AVERAGING aggregation function under the Gradient Ascent attack. This data poisoning manages to slow down the training but not degrade the model's performance to random levels. Error bars represent the standard error.

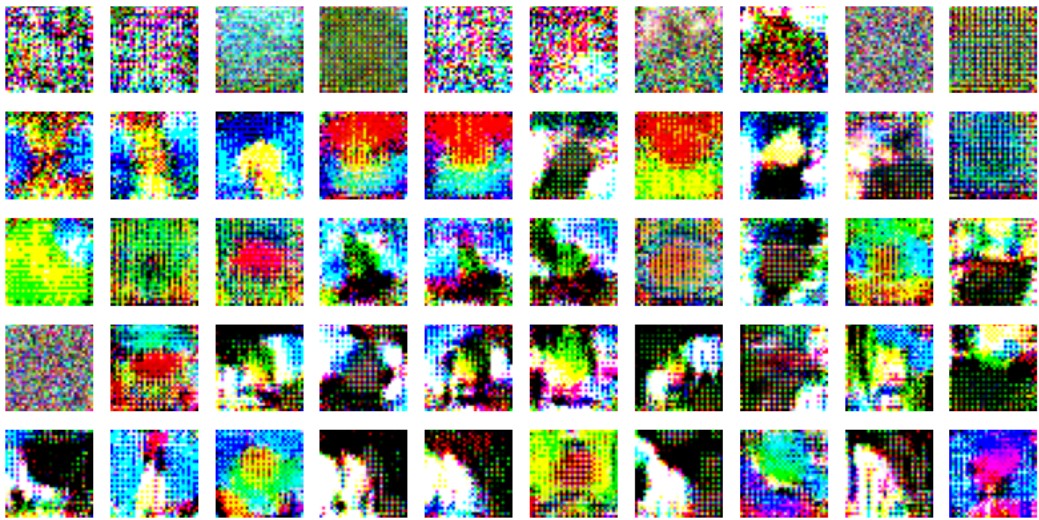

Figure 7: 50 of the first 500 poisons crafted in the (SGD & AVERAGING, Little is Enough, $\alpha = 0.01$) setting.

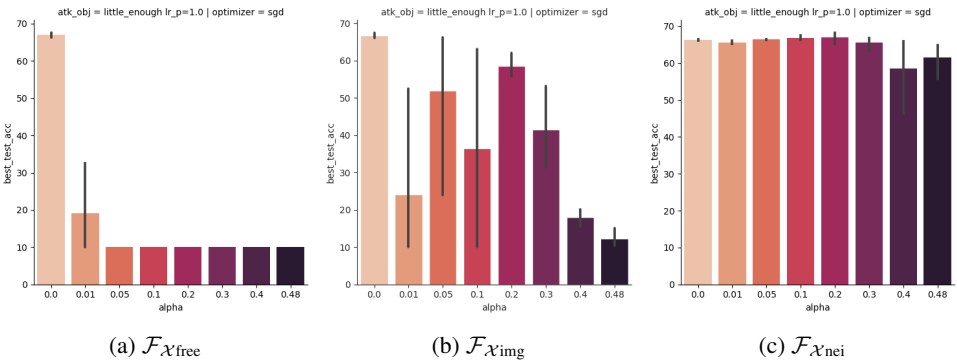

(a) $\mathcal{F}_{\mathcal{X}\mathrm{free}}$        (b) $\mathcal{F}_{\mathcal{X}\mathrm{img}}$        (c) $\mathcal{F}_{\mathcal{X}\mathrm{nei}}$

Figure 8: Best test accuracies for the SGD; AVERAGING setting under the Little is Enough attack for different feasible sets.

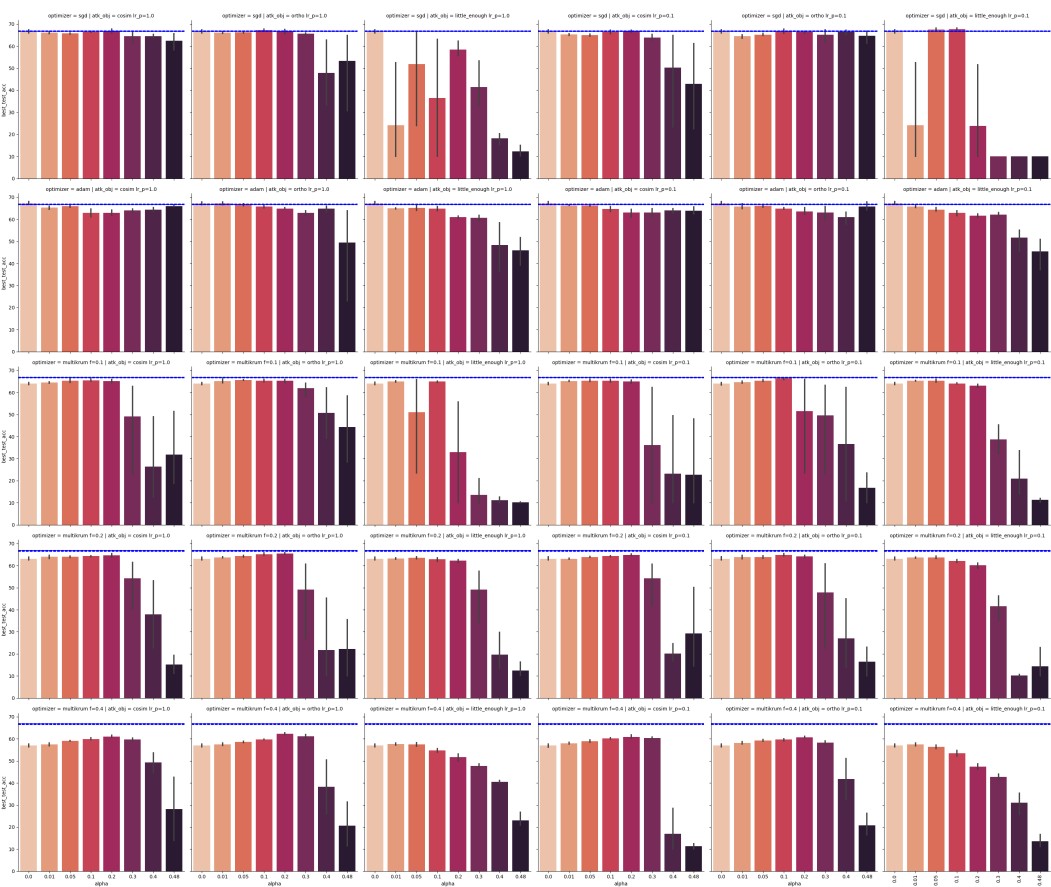

Figure 9: Test accuracy of the model which achieved the best validation accuracy during the training procedure. The blue line is the test accuracy obtained with SGD and no attack.

