# OpenReview forum: "From gradient attacks to data poisoning"
_ICLR.cc/2024/Conference — ICLR 2024 Conference Withdrawn Submission_

### Official Review · Reviewer_Kx6Q · 2023-10-30

**Soundness:** 2 fair
**Presentation:** 1 poor
**Contribution:** 2 fair
**Rating:** 5
**Confidence:** 4

**Summary:**

Assuming that the adversary can append poisoning samples into the minibatch at each iteration, this study investigates the realization of gradient attack through data poisoning. Empirical investigation demonstrates that data poisoning can simulate several types of gradient availability attacks.

**Strengths:**

This study tries to make a connection between gradient attacks and data poisoning. To this end, this study introduces a novel threat model and an attack algorithm under the threat model.

**Weaknesses:**

Overall, the paper lacks proper notations and definitions to represent the authors' ideas.
For example, Section 4 introduces an attack method, while the description of the algorithm lacks proper notations and definitions, and it is hard for the reviewer to understand it.

**Questions:**

Overall, the paper lacks proper notations and definitions to represent the authors' ideas. For this reason, the reviewer could not properly understand

Threat model: The proposed threat model allows the adversary to append poisoning samples into the minibatch at each iteration. In the federated learning setting, this threat model would be justified because the adversary can take part in the federated learning protocol and can inject any samples into its batch. However, in the centralized setting, the reviewer could not find a good reason why this setting is worth consideration.

Poisoning scheme: I could not understand the meaning of eq. 1. In eq. 1, function f_p is minimized w.r.t. S in \mathcal{F}. What is f_p? This is called the "poisoning function" in the manuscript, but I could not find the definition. Also, I could not understand what S \in \mathcal{F} means.

Technical novelty: The following can be incorrect due to my lack of understanding of the proposed idea. In my understanding, the proposed method is to find poisoning samples that cause a specific type of gradient attack. If the technical contribution is based on this technique, I think its novelty is limited.

In page 4: The authors defined the learner's goal to achieve the lowest loss on a test set, which is wrong. The goal would be to attain the lowest generalization error, and the mean is to minimize the training error. The test error is simply used as an estimator of the generalization error, which the learner should not minimize.

---

### Official Review · Reviewer_AvRP · 2023-10-30

**Soundness:** 1 poor
**Presentation:** 1 poor
**Contribution:** 1 poor
**Rating:** 1
**Confidence:** 5

**Summary:**

The paper introduces a method for performing data poisoning attacks that mimic gradient attacks. The effect of such attacks is evaluated in federated learning settings.

**Strengths:**

+ Poisoning attacks are a relevant topic, and they can have an impact on different ML applications.

**Weaknesses:**

- The paper exceeds the page limit (this could be, perhaps, a reason for desk rejection).
- In Section 3.1 the optimization problem is not well formulated. The parameters of the model depend on the training data and the training objective. But this dependency is not show explicitly in the formulation provided by the authors. It is unclear the problem they are trying to solve.
- The settings and the threat model are unclear: the authors mixed different poisoning attacks strategies from centralized and federated machine learning.
- Equation 1, which is the basis for the entire attack is not well explained and justified. It is unclear how this attack compares to the formulation of optimal attack strategies (relying on bilevel optimization), gradient-matching strategies like Witches’ Brew, or different model poisoning attacks in federated learning. The novelty and contributions of the paper are also unclear.

**Questions:**

It is difficult to understand what is the scope of the paper and the settings where this attack can be applied. There are some aspects that require clarification to understand better the contribution and assess the soundness of the approach. For example:
+ What is the novelty of the paper?
+ Can the authors clarify the threat model?
+ Can the authors explain and justify equation 1?
+ Can the authors explain the equation in Section 3.1?

---

### Official Review · Reviewer_jPwq · 2023-11-01

**Soundness:** 1 poor
**Presentation:** 1 poor
**Contribution:** 2 fair
**Rating:** 3
**Confidence:** 5

**Summary:**

The paper proposes a data poisoning attack (probably for federated learning) where the attacker can insert poisonous data in a training batch at each training step. The attacker uses auxiliary dataset to approximate a good gradient and computes poisonous data s.t. the corresponding poisonous gradient can reduce the accuracy of the resulting model. To compute poisonous data, the attacker uses an optimization based on existing model poisoning attacks. Experiments show mixed results, i.e., in some case the attack succeeds and in some cases it fails.

**Strengths:**

- Data poisoning is an important problem for federated learning

**Weaknesses:**

- Threat model is not clear but looks like might be very impractical
- Paper is difficult to read and understand.
- Evaluation set up is not clear and results are also not promising/clear.

**Questions:**

- It seems tough for an attacker to submit data in each batch/round of FL as the threat model; also note that this would be impossible in centralized learning.
- I feel the paper requires more work at least in terms of presentation. I see that all the model poisoning attacks mentioned in Sec. 4.1 are for federated learning but the paper never clarifies the learning setting.
- In Sec 4, I am not sure how LIE objective can be used for the attacks: how can you minimize || g^p - g^a || to get an attack? I am assuming z^{max}.\sigma is a constant. Also what is g^p in this equation?
- Note about LIE: it is an aggregation rule independent attack unlike what Sec 4.1 says.
- In method: what is the intuition behind using attacks in Sec 4.1? If I understand correctly, attacker tries to craft poisonous data s.t. it gives a poisonous gradient that minimizes a particular objective. But the objectives considered here are not that of state-of-the-art model poisoning attacks against FL [1]
- Also Sim_{cos} is not defined anywhere before Table 1
- Results of attack are not great as the paper says, so I am not sure what is the utility of these data poisoning attacks? Why have you not compared with any baseline SOTA data poisoning availability attacks, e.g., that by Shejwalkar et al. (2021)?



[1] Shejwalkar and Houmansadr, Manipulating the byzantine: Optimizing model poisoning attacks and defenses for federated learning, NDSS 2021

---

### Official Review · Reviewer_vYfW · 2023-11-03

**Soundness:** 2 fair
**Presentation:** 2 fair
**Contribution:** 2 fair
**Rating:** 3
**Confidence:** 4

**Summary:**

This paper studies the problem of conducting availability data poisoning attacks against centralized learning framework but still resembles some distributed learning framework, such as robust aggregation.

**Strengths:**

1. The main idea of demonstrating the possible equivalence between data poisoning attacks and gradient attacks under non-convex settings is interesting.
2. The proposed attack achieves good performance in certain settings.

**Weaknesses:**

1. I do not understand how practical the assumed threat model is: the attacker can continuously monitor the training process of the model in each iteration and correspondingly inject the poisoning points. If there is such an application scenario, the authors should clearly articulate the possible use cases. Simply arguing from the perspective of worst case scenario is not sufficient, as no system can withstand an attack under unrealistically strong assumptions. As an attack paper, it is expected to make as less assumption of an assumption as possible to the attacker capability.
2. The experiments using poisoning ratio as high as 0.48 is not realistic. If an attacker can continuously inject different poisoning points in each iteration using as high as 48% of poisoning ratio for each mini batch, I would expect the model to completely fail in the end, but is simply not the case with the experiments, indicating that the attack indeed can be significantly improved. Related to this, the designed poisoning attacks also achieve much performance at a much higher poisoning ratio in some settings.
3. Some of the conclusions are straightforward. For example, using a restricted constraint set to generate the poisoning points can impact attack performance because the valid data points generated may not be able to generate the desired poisoned gradients.

**Questions:**

1. Is there an application scenario where the proposed attack can be a threat?
2. what the data poisoning attacks perform worst at higher poisoning ratios?

---

### Author Response · Authors · 2023-11-23

We wanted to thank the reviewers for their time and constructive comments. It appears that we did not write and present the paper in a way that allowed our point to be understood. We will make sure our point is formulated more clearly on the next iteration.

The goal of this paper was to show that even though data poisoning operates on a much stricter space (as gradients can populate $\mathbb{R}^{d}$ with $d$ the model size whereas data poisons require to respect a certain data encoding), by allowing a data poisoning procedure to operate on a similar threat model as gradient attacks (with regards to the attacker's knowledge and interactions with the model), we can improve their effectiveness. By doing so, a data poisoning on the availability which degrades the performances of the model to random-level becomes possible in non-convex settings. Going through the extra step of finding a data point that corresponds to a gradient attack is not strictly limiting the potential damage of data poisoning. We have not seen such results in the literature to the best of our knowledge.

This result suggests that the gap between gradient attacks and data poisoning attacks might be tighter than what is commonly believed in the research community, starting with [1] which states: "Such attacks are known as model poisoning attacks and are shown to be substantially more impactful on FL than traditional data poisoning attacks.” We hoped that this work would help better understand the difference of effectiveness between the two types of attacks.